# A Retrospective Analysis from Western Romania Comparing the Treatment and Survivability of p16-Positive versus p16-Negative Oropharyngeal Cancer

**DOI:** 10.3390/cancers16050945

**Published:** 2024-02-26

**Authors:** Alexandru Chioreanu, Nicolae Constatin Balica, Cristian Ion Mot, Radmila Bugari, Raluca Morar, Flavia Baderca, Teodora Daniela Marti, Casiana Boru, Cecilia Roberta Avram, Sorin Dema, Dan Dumitru Vulcanescu, Delia Ioana Horhat

**Affiliations:** 1Department of Otorhinolaryngology, “Victor Babes” University of Medicine and Pharmacy, 300041 Timișoara, Romania; chioreanu.alexandru@umft.ro (A.C.); balica@umft.ro (N.C.B.); cristianmotz@yahoo.com (C.I.M.); raluca.morar@umft.ro (R.M.); horhat.delia@umft.ro (D.I.H.); 2Otorhinolaryngology Clinic, Emergency City Hospital, 300054 Timișoara, Romania; 3OftalmoSensory-Tumor Research Center-ORL (EYE-ENT), “Victor Babes” University of Medicine and Pharmacy, 300041 Timișoara, Romania; 4Department of Otorhinolaringology, “Vasile Goldis” Western University of Arad, 310045 Arad, Romania; bugari.radmila@student.uvvg.ro; 5Department of Microscopic Morphology/Histology, Angiogenesis Research Center, “Victor Babes” University of Medicine and Pharmacy, 300041 Timișoara, Romania; baderca.flavia@umft.ro; 6Service of Pathology, Emergency City Hospital, 300254 Timișoara, Romania; 7Department of Medicine, “Vasile Goldis” University of Medicine and Pharmacy, 310414 Arad, Romania; marti.teodora@uvvg.ro (T.D.M.); boru.casiana@uvvg.ro (C.B.); 8Department of Microbiology, Emergency County Hospital, 310037 Arad, Romania; 9Department of Residential Training and Post-University Courses, “Vasile Goldis” Western University, 310414 Arad, Romania; 10Discipline of Radiology, “Victor Babes” University of Medicine and Pharmacy, Eftimie Murgu Square 2, 300041 Timișoara, Romania; sorin.dema@umft.ro; 11Department of Microbiology, “Victor Babes” University of Medicine and Pharmacy, 300041 Timișoara, Romania; dan.vulcanescu@umft.ro; 12Multidisciplinary Research Center on Antimicrobial Resistance (MULTI-REZ), Department of Microbiology, “Victor Babes” University of Medicine and Pharmacy, 300041 Timișoara, Romania

**Keywords:** oropharyngeal cell carcinoma, p16 overexpression, radiotherapy, HPV

## Abstract

**Simple Summary:**

The recent increase in oropharyngeal cancer cases in Western countries prompted this research, which focuses on evaluating the role of human papillomavirus (HPV) in the incidence of this cancer type in Romania. This study employed p16 overexpression as a biomarker to distinguish HPV-related oropharyngeal cancers from those attributed to other etiologies, like smoking and alcohol consumption. Comparative analysis of treatment responses revealed a markedly improved prognosis in patients with p16-positive oropharyngeal cancers, underscoring the significance of HPV in the pathogenesis and clinical outcomes of these malignancies.

**Abstract:**

Background: Oropharyngeal cancer is a global health concern due to its multifaceted nature. Recent molecular studies have linked p16 overexpression, associated with the human papillomavirus, to oropharyngeal cancer and its prognostic implications. Materials and Methods: This retrospective study in Western Romania examined 60 patients, categorizing them based on p16 biomarker status: 28 were p16 positive, and 32 were p16 negative. Statistical tests, including Fisher’s exact and chi2, were used for analysis. Results: Patients with p16-positive oropharyngeal cancer exhibited a better prognosis (3-year survival, *p* = 0.0477; midtreatment, *p* = 0.0349) and reported lower alcohol (*p* = 0.0046) and tobacco (*p* < 0.0001) use. Conclusions: The study highlights the importance of p16 testing in oropharyngeal carcinoma diagnosis. It suggests modifying treatment approaches based on p16 status and underscores the differing prognoses associated with p16-positive and p16-negative cases.

## 1. Introduction

Oropharyngeal cancer (OPC) is a malignancy that affects the oral cavity and pharyngeal region [1]. In recent decades, research has revealed a significant association between *human papillomavirus* (HPV) infection and the development of oropharyngeal cancer [2,3,4]. HPV represents a family of sexually transmitted viruses that affect the epithelia of the mucosa and skin [5].

HPV has been identified as the main risk factor for the development of cervical cancer [6,7]. Thus, research has revealed a significant increase in the prevalence of HPV in oropharyngeal cancer cases, drawing attention to the link between this viral infection and the development of this type of cancer.

This association between HPV and oropharyngeal cancer has brought new insights into the understanding of the mechanisms of development of this type of cancer and has led to important changes in the management and treatment of patients. For example, an increased prevalence of HPV 16 infections is observed in oropharyngeal cancer, which may influence screening, diagnostic, and therapeutic strategies [8,9,10].

The p16 protein is a specific protein that has an important role in cell cycle regulation. It plays a crucial role in suppressing uncontrolled cell growth and preventing tumor formation. Normally, p16 levels are low in healthy cells but accumulate in the case of genetic changes or viral infections, such as *human papillomavirus* infection [8,11,12].

In relation to HPV and cancer, the protein p16 has a significant role. HPV infection causes an increase in p16 expression in infected cells. Thus, increased levels of p16 can be used as a marker for the presence of HPV infection in tumor tissue. Immunohistochemical testing of p16 can be used to identify HPV activity in oropharyngeal cancer [12,13].

Romania is characterized by one of the highest alcohol consumption rates in Europe, and although it has seen a significant reduction in tobacco use, it still faces one of the EU’s highest cancer mortality rates [14]. Adjacent to Romania, Hungary reports the EU’s highest incidence of oral and oropharyngeal cancer in males [15]. Demographically similar and sharing over 800 years of common history until their separation post-World War I, Hungary and Western Romania present a unique multiethnic composition, including Romanians, Hungarians, Serbs, and Germans, among others. The prevalent lifestyle choices, such as consuming smoked meats, high alcohol intake, and tobacco use, potentially impact health outcomes significantly. This positions Western Romania as a critical area for examining the confluence of various factors—ethnic, occupational, dietary, and smoking habits.

The purpose of this article was to examine in depth the relationship between p16 overexpression and oropharyngeal cancer, focusing mainly on its treatment. By analyzing the results of recent studies and research in the field, the importance of determining the p16 protein in any patients with oropharyngeal carcinoma, the modification of surgical, radiotherapeutic, and chemotherapeutic treatment in western Romania, and also the different prognosis of p16-positive oropharyngeal cancer compared to p16-negative oropharyngeal carcinoma will be highlighted in this geographical region.

## 2. Materials and Methods

This was a retrospective analysis comparing clinical outcomes between patients diagnosed with oropharyngeal cancer based on their p16 status. The study followed patients who were admitted between 2015 and 2019 and were able to be followed up for the study of 3-year survivability. Two distinct groups were identified for the study: one consisting of 28 patients who tested positive for the p16 biomarker and another consisting of 32 patients who tested negative.

Medical records were exhaustively reviewed to gather pertinent clinical information, treatment protocols, and patient outcomes. Specific data points, such as age, gender, stage at diagnosis, types of treatment received, alcohol consumption, smoking, and follow-up details, were methodically extracted and prepared for comparative analysis. The inclusion criteria were adult patients with oropharyngeal cancer. Exclusion was based on a lack of complete data sets, patients with multiple primary neoplasms, secondary cancers that received chemotherapy, significant comorbidities that pose additional risks for the treatment like final stages of cardiac, hepatic, pulmonary, or kidney disease, and treatment regimens that did not follow the selected procedure, as explained in the following paragraphs.

In most cases, biopsies were performed under endoscopic control, while for palatine tonsillar cases with suspicion of malignancy, excision within safety margins was performed directly, as described in the local protocols. Other gathered medical information included comorbidities, lifestyle, and treatments undertaken. Secondly, an imagistic examination was undertaken: CT or MRI of the head, neck, and in some cases thorax. Following the confirmation of oropharyngeal carcinoma, p16 overexpression was identified. Patients began treatment; they were sent every week for clinical and endoscopically evaluation of tumor site and lymph nodes. Also, follow-up was performed 3, 6, 12, 24, and 32 months after the finalization of the treatment.

Treatment consisted mainly of radiotherapy, with a dose of 35 × 70 Gy on the tumor site and 20 × 50 Gy on the lymph node areas, all of which were chemotherapeutically potentiated with cisplatin. The patients were followed up weekly during the radiotherapy sessions for tumoral evolution, through endoscopy, pharyngeal examination, and inspection and palpation of the lymph node areas.

Data were collected according to the requirements of the EU Regulation 679/2016 on the protection of individuals regarding the processing of personal data and their free circulation. The study was conducted with the approval of the Ethics Commission of the Timisoara Municipal Emergency Clinical Hospital No. E-5272/9 September 2021.

### Statistical Analysis

The sample size was calculated using the G*Power software (v 3.1.9.6, Heinrich Heine Universitat, Dusseldorf, Germany), using an a priori test to calculate the minimum sample size for a large effect size (w = 0.5), a power of 80%, and an alpha of 0.05. The result indicated that a minimum of 52 patients were needed.

In order to analyze the association between categorical variables in this study, both the chi-square test and Fisher’s exact test were performed [16]. The chi-square test was utilized when the expected frequencies in each cell of the contingency table were sufficiently large (age, tumor site, nodal stage, midtreatment evaluation). Where the expected cell frequencies were small (sex, smoking habits, alcohol abuse, survivability), thereby violating the assumptions of the chi-square test, the Fisher’s exact test was used to ensure a more accurate evaluation of the association, while survival analysis was carried out using the Kaplan–Meier test. All statistical analysis was performed using the MedCalc Statistical Software, version 20.218 (MedCalc Software bv, Ostend, Belgium).

## 3. Results

The 60 patients were divided into two groups, group 1 being made up of p16-positive patients, comprising a total of 28 patients, and group 2 being made up of 32 p16-negative patients. The patients underwent treatment at the Timisoara Municipal Emergency Clinical Hospital, being admitted to the ENT department between 2015 and 2019 with a 3-year follow-up. All demographic data can be observed in Table 1.

The study was stratified based on p16 status into two distinct groups: p16-negative (p16−) and p16-positive (p16+) patients. Of the 60 participants enrolled, the p16− group comprised 32 patients, or 53.3% of the total sample, whereas the p16+ group consisted of 28 patients, or 46.67% of the total sample. Diagnostic or investigative biopsy procedures were conducted on all participants. Additionally, a specific subset, which consisted of four patients from each group (representing 12.5% of the p16− group and 14.29% of the p16+ group), underwent tumoral excision surgeries. These patients have been presented to the oncological committee, and their cases were discussed by physicians from each of the following specialties: ENT, oncology, radiology, radiotherapy, and anatomopathology. Next, the patients received the same protocol regarding radiotherapy similar to the patients that did not undergo surgery.

In terms of age demographics, the average age of patients in the p16− group was determined to be 59.87 years, while for the p16+ group, the average age was 56.28 years. Differences in the mean age between the two groups may warrant further statistical scrutiny. Furthermore, the study’s gender distribution indicated a predominant male participation in both groups. Specifically, out of the 32 patients in the p16− group, 3 were female (9.4%) and 29 were male (90.6%). The p16+ group had 6 female participants (21.4%) and 22 male participants (78.6%).

Regarding the age groups, the study found that for ages 30–40, both groups were nearly the same, with two patients from each group in this age range. For ages 41–50, the p16-positive group had more people, with nine (or 28.1%), while the p16− group had five (or 17.9%). In the 51–60 age bracket, the p16+ group had three people (or 9.4%), but the p16-negative group had slightly more, with five people (or 17.9%). The biggest age group was 61–70, where the p16+ group had 13 people (or 40.6%); meanwhile, the p16-negative group had even more, with 16 people (or 57.1%). Lastly, in the 71–80 age group, patients with p16-positive overexpression only had one person (or 3.1%), and the negative group had four people (or 14.3%).

Initial findings suggest potential variations in age and gender distributions between the p16− and p16+ groups, as compared to other studies where patients that had a p16+ overexpression were younger. Out of 51 males, 22 (or 78.6%) were in the p16+ group, while 29 (or 90.6%) were in the p16− group. Of the nine females in the study, six (or 21.4%) were in the p16-positive category, and three (or 9.4%) were in the p16-negative group. There is clearly a prevalence of OPC in the male population, while a higher percentage of females had a positive p16 overexpression, showcasing the possible implications of HPV in the female population.

The association between smoking and oropharyngeal cancer is well established [17,18]. Within the cohort of 41 individuals who identified as smokers, 11 (or 39.3%) were p16+, whereas a notably larger proportion, 30 (or 93.8%), were p16 negative.

Alcohol abuse alongside smoking is always strongly related to the development of oral and oropharyngeal cancer. In this study, among the individuals identified as engaging in alcohol abuse, out of 27 in total, 7 (25.9%) were from the p16+ group, whereas a considerably larger portion, 20 (74.1%), were from the p16 negative. These data indicate a potential correlation between the p16− group and a higher likelihood of alcohol abuse.

p16+ and p16− OPC has been found in almost all four oropharyngeal anatomical sites, as follows: the p16− group manifested a minor prevalence with one case, making up 3.1% of its total cohort, while p16+ had none. All data regarding tumoral localization and extension are provided in Table 2.

The palatine tonsils are by far the most common place where the OPC was found, indifferent of the two groups. A substantial 75.0% of the p16+ group, equivalent to 21 individuals, were diagnosed with tumors at this site. In contrast, the p16− group demonstrated a lower, yet notable, frequency with 19 cases, which translates to 31.3% of its members. They comprised over 66.0% of total carcinomas.

For tumors located at the lingual tonsils, patients with p16-positive overexpression reported five individuals, or 17.9% of its members, with such tumors. The p16− group, on the other hand, showed a higher predilection with 10 individuals affected, amounting to 31.25% of the group. Lastly, when considering the soft palate as the tumor site, both groups exhibited a somewhat comparable distribution. Similar proportions within both groups were reported to have been located on the soft palate (four patients in total). However, no statistically significant differences were observed.

Regarding staging, there were no patients with T1 in the p16+ cohort. In contrast, the p16− group registered a small percentage, with two individuals, which amounts to 6.3% of its group. Delving into the T2 stage, the 16+ group saw representation from four individuals, making up 14.3% of the cohort. Meanwhile, the p16− group had a slightly diminished occurrence with three individuals, or 9.4% of its members, falling under this stage. For the T3 tumor stage, the p16+ group documented five patients, or 17.9% of its total members. This is juxtaposed with the p16− group, where seven individuals, translating to 21.9% of its cohort, had a T3 tumor. Finally, in the T4 stage, a predominant share of both groups was found. In the p16+ group, there were 19 patients. Similarly, the p16− group was not far behind, with 20 individuals, amounting to 62.5% of the group, being diagnosed within this stage. In order to properly analyze the data, Fisher’s test was used; hence, the T1 and T2 stages and the T3 and T4 stages, respectively, were combined. However, no statistically significant differences were observed.

When evaluating the nodal stages in relation to p16 status, the data revealed the following aspects:

In the Nx stage, which typically denotes the inability to evaluate regional lymph nodes, the p16+ group presented three cases, accounting for 33.3% of the total cases under this stage. In comparison, the p16− group had a larger representation with six cases, contributing 66.7% to the total Nx cases.

For the N0 stage, indicating no regional lymph node involvement, the p16+ group exhibited four cases, making up 57.1% of the total cases for this stage. The p16− group had three cases, which form 42.9% of the total N0 cases.

The N1 stage, which describes the involvement of a single lymph node, represented two cases in the p16+ group, or 22.2% of the total. The p16- group showcased a higher frequency with seven cases, equating to 77.8% of the total N1 cases. When observing the N2a stage, both groups exhibited a balanced distribution. Both the p16+ and p16− groups had three cases each, representing 50.0% of the total N2a cases. In the N2b stage, where multiple lymph nodes on the same side of the primary tumor are involved, the p16+ group had a significant presence with 10 cases, translating to 83.3% of the total cases. The p16− group reported two cases, which is 16.7% of the total N2b cases.

The N2c stage, indicating lymph nodes on both sides of the neck are involved, was exhibited in the p16+ group in six cases, or 54.6% of the total. The p16− group was closely matched with five cases, making up 45.4% of the N2c cases. Interestingly, the N3a stage saw no representation from either group, with both having no cases.

Lastly, in the N3b stage, a distinct pattern emerged where the p16+ group had no representation, while the p16− group had a complete presence with all six cases or 100.0% of the total. In order to properly analyze the data, Fisher’s test was used; hence, the Nx, N0, and N1 stages and the N2 and N3 stages, respectively, were combined. However, no statistically significant differences were observed.

The relationship between p16 status and 3-year survivability offers insightful conclusions about patient outcomes. Among individuals diagnosed, those in the p16+ category displayed a notably high 3-year survivability rate, with 23 out of 28, or 82.1%, surviving past this milestone. This stands in stark contrast to the p16− group, where only 18 out of 32, or 56.3%, achieved the same 3-year threshold. This was statistically significant, according to data on survivability and treatment, which are presented in Table 3.

Additionally, the inverse relation, namely the cases that did not reach the 3-year survivability mark, further emphasizes this disparity. In the p16+ group, only 5 out of 28, which translates to 17.86%, did not survive past 3 years. Meanwhile, the p16− cohort displayed a more pronounced number of nonsurvivors, tallying 14 out of 32, or 43.75% of their total count. Survivability was also assessed using the Kaplan–Meier analysis, as can be observed in Figure 1. The Kaplan–Meyer probability curve of mortality p16+ and p16− patients showed different, statistically significant risks (log-rank *p*-value = 0.0328).

The last criterion of the study was the treatment response during the radiotherapy sessions, so there was a clinical evaluation after 15 courses of radiation that involved an inspection of the head and neck via nasal, pharyngeal, and laryngeal endoscopy. During the midtreatment clinical evaluation for patients with different p16 statuses, distinct treatment response patterns emerged.

Patients exhibiting a positive p16 status demonstrated a notably higher incidence of complete regression compared to their p16− counterparts. Specifically, half (50%) of the p16+ patients showed complete regression; thus, there was no tumoral presence in the clinical examinations, this does not necessarily mean that there were no microscopic lesions left on the tumoral bed. Within the p16− group, only approximately 21.9% of patients had a complete regression. When considering both groups together, 35% of the total cohort exhibited a complete regression. This suggests that the p16+ status might be indicative of a more favorable response to treatment, at least in terms of complete regression.

However, the pattern shifts when observing partial regression rates. The p16− group had a higher proportion of patients (59.4%) with partial regression compared to 39.3% in the p16+ group. This indicates that while a greater proportion of p16− patients showed some response to the treatment, it was not as definitive as the complete regression seen in the p16+ group. Overall, 50% of the combined cohort exhibited partial regression.

Regarding nonresponders, 10.7% of the p16+ patients showed no response to the treatment, lower than the 18.8% observed in the p16− group. Across the entire study population, 15% did not show any discernible response during the half period of radiotherapy. In order to properly analyze the data, Fisher’s test was used; hence, the partial and no response categories were combined. Statistically significant differences were observed at *p* = 0.0227.

## 4. Discussion

Human papillomavirus (HPV) is among the most widespread viral infections and has been associated with several pathologies. It has a diverse group of viruses, ranging from mostly benign and low-risk subtypes like HPV-11 and HPV-6 that can also be involved in laryngeal pathologies to particularly high-risk subtypes like HPV-16 and HPV-18, which are linked to oropharyngeal cancer [18].

Smoking and alcohol were the primary culprits for oropharyngeal cancers before the implications of HPV in carcinogenesis were discovered. However, as tobacco consumption patterns have changed, there has been a paradigm shift in the epidemiology of this disease [6]. Over the past few decades, there has been an undeniable rise in oropharyngeal cancers attributed to HPV, especially in developed nations [10,19,20]. Association with smoking, alcohol consumption, and poor local hygiene has proven to enhance the potential carcinogenic effect. Nowadays, it is not uncommon for clinicians to encounter young and presumably healthy patients diagnosed with oropharyngeal cancer located specifically at the palatine tonsils or the lingual tonsils [12,21,22].

HPV’s carcinogenic potential is driven by its E6 and E7 oncoproteins [21,23,24,25]. These proteins provoke uncontrolled cell proliferation. The E6 protein targets the p53 tumor suppressor protein for degradation, while E7 binds and inactivates the retinoblastoma (Rb) protein, starting a process of cellular immortalization and transformation [3,26,27]. Meanwhile, the p16 protein is of utmost importance from a clinical point of view [3,8,10]. p16 acts as a cell cycle regulator, inhibiting cyclin-dependent kinases. However, in HPV-transformed cells; due to the inactivation of Rb by HPV’s E7 protein, there is an overcompensation leading to overexpression of p16. An elevated presence of p16 in oropharyngeal tumors has become almost synonymous with HPV involvement, making it a surrogate marker. Also, homeobox genes responsive to *KDM6A* and *KDM6B* exhibit significantly elevated expression levels, indicating that HPV16 E7 induces reprogramming of host epithelial cells. Notably, these effects persist irrespective of E7’s capacity to inhibit the retinoblastoma tumor suppressor protein. Crucially, the reversal of these effects upon silencing E7 expression suggests that this pathway could hold prognostic and/or therapeutic significance [28].

The induction of *p16(INK4A)* expression by oncogenic stress initiates cellular senescence via activation of the retinoblastoma tumor suppressor. The evasion of oncogene-induced senescence is a critical stage in cancer development, with numerous tumors exhibiting the loss of *p16(INK4A)* activity through mutation or epigenetic silencing [29]. *p16(INK4a)*-dependent growth arrest was also described, as RAS disrupts PcG-mediated repression of the *INK4a/ARF* tumor suppressor locus. This is achieved through the activation of the *H3K27 demethylase JMJD3* and the suppression of the *methyltransferase EZH2*. In human fibroblasts, *JMJD3* specifically activates *p16(INK4a)*, resulting in growth arrest, while *ARF* remains unaffected [30]. The reduction in *p16(INK4A)* gene expression in three distinct cervical carcinoma cell lines led to a decrease in the expression of the E7 oncogene. This implies the existence of a positive feedback loop involving E7 and *p16(INK4A)* [31].

Thus, this study highlights the potential differential treatment responses in oropharyngeal cancer patients based on their p16 status. The p16+ patients seem to have a more pronounced rate of complete regression, while a significant proportion of p16− patients showed partial responses. The data underscore the importance of considering molecular markers like p16 in tailoring treatment strategies and predicting outcomes for oropharyngeal cancer patients. What is more compelling is that the biological behavior of HPV-positive oropharyngeal cancers is distinctively different from their HPV-negative counterparts. Patients with HPV-positive tumors generally have a more favorable prognosis. Their cancers, for reasons still under investigation, respond better to both radiation and chemotherapy.

p16 overexpression is not the most specific analysis regarding the presence of HPV in any type of carcinoma, although it is highly sensitive to the viral infection, but it remains much more cost-efficient than Southern blots, dot blot, in situ hybridization (ISH), polymerase chain reaction, or other methods. Some studies have shown a specificity of more than 80%, with rates of misidentification as low as 5–20% [32].

As this study reveals, patients with p16+ overexpression tend to have a more favorable outcome regarding the speed of regression, low relapses, and high survivability, which is not the case for p16− patients who are likely HPV negative and thus fall into the smoking/alcohol consumption criteria for OPC. Women tend to have a lower incidence of OPC, especially p16−, possibly because they have a healthier lifestyle regarding smoking, oral hygiene, and alcohol consumption. Furthermore, some articles suggest that women are protected by hormonal status [33]. Additionally, while most other articles suggest a higher prevalence of young adults developing p16+ OPC [21], Eastern Europe has not yet presented the same prevalence as Western Europe and the American continent, with some places having over 70% of OPC being HPV positive, compared to this study with under 50% [5].

Smoking and alcohol consumption continue to pose a challenge for local health systems, causing a high number of morbidities, deaths, and draining of resources. This study acknowledges the aspect of a faster and more precise response to radiotherapy, confirming international studies about p16 and HPV-positive oropharyngeal cancer and the need to rethink the radiation protocols for OPC HPV+ [8,10,34,35]. Similar to other organs and different types of cancers, a better understanding of the relationship between inflammation and malignant tumors and the discovery of more oncogenes implicated in the tumorigenesis of these conditions will support the development of new and targeted treatments that will improve patients’ quality of life [36].

Prevention is one of the keys to lowering oropharyngeal cancer rates; thus, national programs like vaccination for HPV for girls and boys should be implemented. Finally, a more aggressive attitude towards tobacco and alcohol consumption should prove to help lower the cancer rates found in p16/HPV- OPC [37].

Studies from Romania regarding research into oropharyngeal cancer and p16 expression are somewhat scarce. One study from the northeastern part of Romania, namely from Iasi, outlined the prevalence of different high-risk subtypes of HPV in their region, with positive correlations for patient survival in regard to the p16 and p53 markers [38,39]. They observed similar behavior patterns regarding tobacco and alcohol consumption and a similar survivorship pattern. In another study from the region, a minute fraction of cases seemed to be influenced by HPV, as indicated by a minimal agreement between the presence of HPV DNA and the HPV RNA or p16 status observed in the analyzed patients [40].

Further studies from our country have talked about other markers in oropharyngeal cancer, such as TNF- α [41], blood group type association [42], MicroRNA-486-5p and MicroRNA-10b-5p [43], as well as the use of micro-Raman and FT-IR spectra of saliva [44].

As such, despite advancements in other regions, research pertaining to p16 expression in oropharyngeal cancer in Romania remains limited. Further comprehensive epidemiological studies would help elucidate the prevalence and trends of p16-positive and -negative oropharyngeal cancer in our country, while in-depth molecular profiling could also improve our understanding of the subject in the region.

### Limitations

As with all studies, our research had some limitations. Firstly, due to the retrospective nature of the research, the accuracy and completeness of data depend on the quality of existing records. As such, patients with incomplete data were eliminated from the study, as mentioned in Section 2. Also, due to economic constraints, HPV in situ testing was not accessible. As such, we acknowledge that p16 is not an absolute test for viral etiology. Another limitation would be that differentiation was focused on TNM staging while not including other methods in our analysis. Due to local protocol, which recommends bilateral tonsillectomy if there is even the slightest suspicion of malignancy in the palatine tonsils, whether unilateral or bilateral, other surgery options for these cases may result in different outcomes.

## 5. Conclusions

This study analyzed p16 overexpression in patients with oropharyngeal carcinoma and examined parameters such as gender, age, tumor site and stage, smoking, alcohol consumption, nodal stage, and, most importantly, survivability at 3 years and treatment evolution. It revealed an almost equal distribution between positive and negative OPC p16 patients, with a larger proportion of males being affected. Additionally, patients with a positive overexpression were younger, possibly implying an HPV infection involving the oropharyngeal space.

Furthermore, the progression and treatment outcomes, as well as survivability at 3 years for patients with p16-positive oropharyngeal carcinoma, were significantly better, surpassing those of patients with p16-negative status. This underscores the potential importance of p16 status as a prognostic indicator in the management and therapeutic strategies for oropharyngeal cancer patients.

In Western Romania, alcohol consumption and smoking continue to be major health concerns. Concurrently, HPV infections are increasingly causing more cancers. The absence of adequate screening and vaccination programs is leading to a worrying trend where HPV-positive oropharyngeal cancers may surpass those attributed to smoking and alcohol in the coming years, similar to the Western world. This shift underscores the urgent need for enhanced public health interventions in these areas.

## Figures and Tables

**Figure 1 cancers-16-00945-f001:**
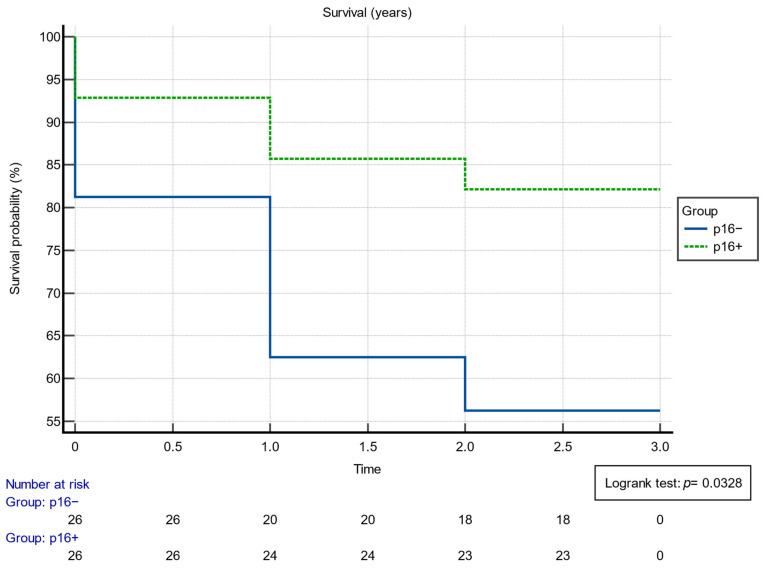
Results of the Kaplan–Meier survival analysis.

**Table 1 cancers-16-00945-t001:** Demographic data.

	p16+ (n = 28)	p16− (n = 32)	Total (n = 60)	*p*-Value
**Age (years)**				**0.4776**
30–40	2(6.3%)	2(7.1%)	4	
41–50	9(28.1%)	5(17.9%)	14	
51–60	3(9.4%)	5(17.9%)	8	
61–70	13(40.6%)	16(57.1%)	29	
>70	1(3.1%)	4(14.3%)	5	
**Gender**				**0.2812**
Male	22(78.6%)	29(90.6%)	51	
Female	6(21.4%)	3(9.4%)	9	
**Smoking**				**<0.0001 ***
Yes	11(39.3%)	30(93.8%)	41	
No	17(60.7%)	2(6.3%)	19	
**Alcohol abuse**				**0.0046 ***
Yes	7(25.9%)	20(74.1%)	27	
No	21(63.6%)	12(36.4%)	33	

*: *p* < 0.05, statistically significant.

**Table 2 cancers-16-00945-t002:** Data regarding tumor localization and extension.

	p16+ (n = 28)	p16− (n = 32)	Total (n = 60)	*p*-Value
**Tumor site**				**0.4733**
Pharyngeal wall	0 (0.0%)	1 (3.1%)	1	
Palatine tonsils	21 (75.0%)	19 (31.3%)	40	
Lingual tonsils	5 (17.9%)	10 (31.3%)	15	
Soft palate	2 (7.1%)	2 (6.3%)	4	
**Tumor stage**				**0.8848**
Low (T1, T2)	4 (44.4%)	5 (55.6%)	2	
High (T3, T4)	24 (47.1%)	27 (52.9%)	7	
**Nodal stage**				**0.1616**
Nx/N0/N1	9 (36.0%)	16 (64.0%)	16	
N2/N3	19 (54.3%)	16 (45.7%)	29	
**Extranodal extension**				**0.0679**
Yes	0 (0.0%)	20 (100.0%)	20	
No	6 (15%)	34 (85.0%)	40	

**Table 3 cancers-16-00945-t003:** Data comparing survivability at 3 years post-radiotherapy and midtreatment clinical evolution for the two p16 groups.

	p16+ (n = 28)	p16− (n = 32)	Total (n = 60)	*p*-Value
**Survivability 3 years**				**0.0314 ***
Yes	23 (82.1%)	18 (56.3%)	41 (68.3%)	
No	5 (17.9%)	14 (43.7%)	19 (31.7%)	
**Midtreatment evaluation**				**0.0227 ***
Complete regression	14 (50.0%)	7 (21.9%)	21 (35.0%)	
Partial or no response	14 (50.0%)	25 (78.1%)	39 (65.0%)	

*: *p* < 0.05, statistically significant.

## Data Availability

Data are available upon request from the corresponding author.

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
