# Peer review of "A Retrospective Analysis from Western Romania Comparing the Treatment and Survivability of p16-Positive versus p16-Negative Oropharyngeal Cancer"

_cancers, 2024, doi:10.3390/cancers16050945_

Round 1

Reviewer 1 Report

Comments and Suggestions for Authors

I have thoroughly reviewed the manuscript titled "A retrospective analysis from Western Romania comparing the treatment and survivability of oropharyngeal cancer with p16 positive versus p16 negative mutations." After careful consideration, I regret to inform you that I do not recommend this paper for publication. The main reasons for this decision are as follows:

1. Lack of Novelty: The study's findings do not contribute significantly to the existing literature on oropharyngeal cancer. The differentiation between p16 positive and negative oropharyngeal cancers and their respective prognoses and treatment responses have been well-documented in numerous studies. Such findings might have been groundbreaking if published around the year 2001, but they no longer offer new insights in the year 2023.

2. Misleading Terminology: for example the use of the term "p16 mutations" in the title is misleading. The overexpression of p16, often associated with HPV, is a well-known phenomenon in oropharyngeal cancer, but it is not typically referred to as a mutation. This choice of words might confuse readers or misrepresent the nature of the study.

3. Complexity of p16 Overexpression: The mechanism behind p16 overexpression is complex and multifactorial. I would advise the authors to deepen their understanding of this subject by consulting key research papers in this field, such as:

   - McLaughlin-Drubin ME, Crum CP, Munger K. "Human papillomavirus E7 oncoprotein induces KDM6A and KDM6B histone demethylase expression and causes epigenetic reprogramming." Proc Natl Acad Sci U S A. 2011.

   - Barradas M, Anderton E, Acosta JC, et al. "Histone demethylase JMJD3 contributes to epigenetic control of INK4a/ARF by oncogenic RAS." Genes Dev. 2009.

   - McLaughlin-Drubin ME, Park D, Munger K. "Tumor suppressor p16INK4A is necessary for survival of cervical carcinoma cell lines." Proc Natl Acad Sci U S A. 2013.

   - Pauck A, Lener B, Hoell M, et al. "Depletion of the cdk inhibitor p16INK4a differentially affects proliferation of established cervical carcinoma cells." J Virol. 2014.

In summary, while the study is methodologically decent, its lack of novelty and certain inaccuracies in terminology and understanding of the p16 pathway make it less suitable for publication in its current form. An updated focus, reflecting recent advancements and a more nuanced understanding of the molecular pathways involved, would be required for this study to add value to the current body of research.

Author Response

Thank you for your comments and suggestions. Please find the response attached.

Reviewer 2 Report

Comments and Suggestions for Authors

This is a very well-planned retrospective study. The background was short, but informative and the aims were clearly formulated. The statistical analyses were appropriate. The text is well written and easy to understand. The tables are clear.

As this paper discusses survival rates and survival times, the authors should perform survival analysis using Kaplan-Meier curves. This is my only suggestion for improvement and therefore mandatory.

Author Response

(The authors gave the same response as above.)

Reviewer 3 Report

Comments and Suggestions for Authors

The manuscript represents a study of oropharyngeal cancer in Romania.

The following are the issues related to the findings in this study:

1) The title needs to change p16 expression.  P16 is tested by IHC for overexpression and not by genomic analysis.

2) P16 negative tumors should be confirmed by HPV in-situ testing. 

P16 is a surrogate marker and is not an absolute test for viral etiology.

3) The histologic variant and the differentiation of tumor tested should be included.  This feature may affect p16 expression scoring.

4) No information on type of resection and margins status are included. This information is critical to assessing local recurrence and post-operative therapy.

5) Type of surgery and post-operative therapy information is missing.

This information should be included.

Author Response

(The authors gave the same response as above.)

Round 2

Reviewer 1 Report

Comments and Suggestions for Authors

Thank you for your detailed and thoughtful responses to the comments I provided during the review of your manuscript. I appreciate the efforts you have made to address the concerns raised, particularly in refining the manuscript's terminology and enhancing its clarity and value through the addition of a Kaplan-Meier survival analysis.

I acknowledge and value the unique contribution of your study in providing insights from the Western Romania patient cohort on p16 positive and negative oropharyngeal cancers. This regional perspective indeed adds a valuable dimension to the existing literature.

However, to ensure that the significance and specificity of your findings are immediately clear to all readers, I would recommend further emphasizing the local context of your study throughout the manuscript. This includes making it explicit in the title, introduction, objectives, and conclusions that the data presented are particularly relevant to Romania. Highlighting this aspect will clarify that these findings address a previously unexplored or underrepresented area in the literature, thereby underlining the novelty and importance of your work within its specific geographical and demographic context.

Author Response

Thank you, modifications were made throughout the document to increase the relevance in regards to other research from our country on this subject. This has been mainly addressed in the new paragraph from the Discussion section and some rework of the Conclusions.

Reviewer 2 Report

Comments and Suggestions for Authors

Thank you very much for adding the survival analysis as suggested!

Author Response

Thank you very much for you support and suggestions!